# DNA vaccination protects mice against Zika virus-induced damage to the testes

Bryan D. Griffin[1,2], Kar Muthumani[3], Bryce M. Warner[2,4], Anna Majer[5], Mable Hagan[1,2], Jonathan Audet[1,2], Derek R. Stein[2,4], Charlene Ranadheera[2,4], Trina Racine[1,6], Marc-Antoine De La Vega[6], Jocelyne Piret[7], Stephanie Kucas[4,8], Kaylie N. Tran[1], Kathy L. Frost[5], Christine De Graff[8], Geoff Soule[1], Leanne Scharikow[4], Jennifer Scott[9], Gordon McTavish[9], Valerie Smid[10], Young K. Park[11], Joel N. Maslow[11], Niranjan Y. Sardesai[12], J. Joseph Kim[12], Xiao-jian Yao[2], Alexander Bello[1,2], Robbin Lindsay[2,4], Guy Boivin[7], Stephanie A. Booth[2,5], Darwyn Kobasa[1,2], Carissa Embury-Hyatt[10], David Safronetz[2,4], David B. Weiner[3] & Gary P. Kobinger[6,13]

Zika virus (ZIKV) is an emerging pathogen causally associated with serious sequelae in fetuses, inducing fetal microcephaly and other neurodevelopment defects. ZIKV is primarily transmitted by mosquitoes, but can persist in human semen and sperm, and sexual transmission has been documented. Moreover, exposure of type-I interferon knockout mice to ZIKV results in severe damage to the testes, epididymis and sperm. Candidate ZIKV vaccines have shown protective efficacy in preclinical studies carried out in animal models, and several vaccines have entered clinical trials. Here, we report that administration of a synthetic DNA vaccine encoding ZIKV pre-membrane and envelope (prME) completely protects mice against ZIKV-associated damage to the testes and sperm and prevents viral persistence in the testes following challenge with a contemporary strain of ZIKV. These data suggest that DNA vaccination merits further investigation as a potential means to reduce ZIKV persistence in the male reproductive tract.

[1] Special Pathogens Program, National Microbiology Laboratory, Public Health Agency of Canada, 1015 Arlington Street, Winnipeg, Manitoba, Canada R3E 3R2. [2] Department of Medical Microbiology and Infectious Diseases, University of Manitoba, 745 Bannatyne Avenue, Winnipeg, Manitoba, Canada R3E 0J9. [3] The Wistar Institute, 3601 Spruce Street, Philadelphia, Pennsylvania 19104, USA. [4] Zoonotic Diseases and Special Pathogens, National Microbiology Laboratory, Public Health Agency of Canada, 1015 Arlington Street, Winnipeg, Manitoba, Canada R3E 3R2. [5] Molecular Pathobiology, National Microbiology Laboratory, Public Health Agency of Canada, 1015 Arlington Street, Winnipeg, Manitoba, Canada MB R3E 3R2. [6] Department of Microbiology and Immunology, Faculty of Medicine, Laval University, 1050 avenue de la Médecine, Québec City, Québec, Canada G1V 0A6. [7] Research Center in Infectious Diseases of the CHU of Québec and Laval University, 2705 boulevard Laurier, Québec City, Quebec, Canada G1V 4G2. [8] Veterinary Technical Services, Public Health Agency of Canada, National Microbiology Laboratory, 1015 Arlington Street, Winnipeg, Manitoba, Canada R3E 3R2. [9] Heartland Fertility & Gynecology Clinic, 701-1661 Portage Avenue, Winnipeg, Manitoba, Canada R3J 3T7. [10] National Centre for Foreign Animal Disease, Canadian Food Inspection Agency, 1015 Arlington Street, Winnipeg, Manitoba, Canada R3E 3M4. [11] GeneOne Life Science Inc., 223 Teheran-Ro, Gangnam-Gu, Seoul, Korea. [12] Inovio Pharmaceuticals Inc., 660 West Germantown Pike, Plymouth Meeting, Pennsylvania 19462, USA. [13] Department of Pathology and Laboratory Medicine, University of Pennsylvania School of Medicine, 3400 Spruce Street Philadelphia, Pennsylvania 19104-4238, USA. Correspondence and requests for materials should be addressed to G.P.K. (email: gary.kobinger@crchudequebec.ulaval.ca).

Zika virus (ZIKV) is an emerging pathogen causally associated with serious sequelae in fetuses, inducing fetal microcephaly and other neurodevelopment defects[1–6]. ZIKV has been shown to persist in human semen[7–9] and sperm[10] up to several months after the onset of symptoms, and male-to-male[11] or male-to-female sexual transmission has been reported[7,12–14]. Consistent with these observations, type I IFN receptor $\alpha$ chain null mice (Ifnar1$^{-/-}$) exposed to ZIKV by the subcutaneous (s.c.) route contain high viral loads in the testes[15,16]. Further, exposure of male Ifnar1$^{-/-}$ mice to ZIKV and exposure of type I IFN-depleted wild-type mice to mouse-adapted ZIKV results in severe damage to the testis, epididymis and sperm with a measurable reduction in fertility[17,18]. Moreover, male rhesus and cynomolgus macaques exposed to ZIKV by the s.c. route have high viral loads in semen, and infected cells can be detected in the prostate, seminal vesicles and testes[19].

Several vaccines have been shown to protect against ZIKV infection in experimental animal models, and have entered clinical trials[20–26]. Our previous work has shown that immunization with a synthetic pre-membrane + envelope (prME) DNA vaccine construct fully protects Ifnar1$^{-/-}$ mice from the weight loss and neurologic disease otherwise associated with a high challenge dose of ZIKV[22]. We therefore sought to conduct a longitudinal study to characterize ZIKV infection of the male reproductive tract in susceptible mice and to evaluate the efficacy of this DNA vaccination in reducing or eliminating the viral burden in these tissues.

In this study, we report that delivery of a synthetic DNA vaccine encoding ZIKV prME prior to a high-dose challenge with a contemporary strain of ZIKV completely protects mice against ZIKV-associated damage to the testes and sperm and prevents persistence of ZIKV in the testes and epididymis. These data suggest that DNA vaccination and investigational therapeutics warrant further examination as a potential means to reduce ZIKV persistence in the male reproductive tract.

## Results

**ZIKV infection results in damage to the testis.** As a model of ZIKV replication and associated pathogenicity, 10 week-old Ifnar1$^{-/-}$ mice were challenged with $1 \times 10^6$ plaque-forming units (p.f.u.) of ZIKV PRVABC59 strain by a s.c. route leading to significant damage to the testes, epididymis and sperm, consistent with recently published findings[17,18]. To investigate vaccine-mediated protection against testicular damages, prime-boost immunizations of Ifnar1$^{-/-}$ mice consisting of intra muscular (i.m.) injection followed by electroporation-mediated delivery of a prME DNA vaccine at 2-week intervals were performed prior to a ZIKV challenge with $1 \times 10^6$ p.f.u. at 10 weeks of age.

Following exposure of unvaccinated Ifnar1$^{-/-}$ mice to ZIKV, ~40% lethality was observed by 8 days post-infection (d.p.i.) with weight loss, neurologic signs and activation of ZIKV-specific T-cells consistent with previous studies (Supplementary Fig. 1). High viral RNA loads ($3 \times 10^7$ to $1 \times 10^{12}$ genome equivalents per gram) were detected in the testes at 7 d.p.i. and persisted at day 21 d.p.i. (Fig. 1a). Infectious virus (up to $3.2 \times 10^8$ TCID$_{50}$ per g) were identified in the majority of mice testes at 21 d.p.i by standard infectious assay (Supplementary Fig. 2a). No ZIKV RNA or infectious virus were detected in uninfected control animals at any point throughout the study. At day 14 post-infection, a decrease in testicle size (Fig. 1b and Supplementary Fig. 2b) and weight (Fig. 1c) were observed in several of the infected mice, which became more pronounced at day 21 with means that were significantly different (two-way analysis of variance (ANOVA) $P < 0.0001$). Microscopic evidence

of damage to the seminiferous epithelium was evident at 7 d.p.i. which may include degeneration of the spermatogenic lineage (Fig. 1d, upper panel). Viral antigen was observed by immunohistochemistry (IHC) multifocally throughout the seminiferous epithelium which includes all layers of maturing spermatogenic cells and Sertoli cells (Fig. 1d, middle panel). In the testis at 14 d.p.i. there were large areas where the normal architecture of the seminiferous tubule was completely replaced by necrotic debris and inflammatory cells, primarily neutrophils (Fig. 1d, upper panel). The connective tissue areas surrounding the tubules were expanded by an infiltrate of mixed inflammatory cells including macrophages, neutrophils and lymphocytes (Fig. 1d, upper panel). Abundant viral antigen was observed in affected seminiferous tubules (Fig. 1d, middle panel). At 21 days the normal architecture was completely effaced with massive loss of seminiferous tubules and replacement by fibrous tissue and inflammatory cells (Fig. 1d, upper panel), and lymphocyte infiltration, including macrophages, was observed (Supplementary Fig. 2c,d). Abundant viral antigen was observed within remaining affected seminiferous tubules (Fig. 1d, middle panel). Through 7, 14 and 21 d.p.i. there was increasing evidence of cell death via the mechanism of apoptosis as assessed by TUNEL staining (Fig. 1d, lower panels). Scattered necrosis of epithelial cells was noted in the epididymis at 7 d.p.i. with abundant necrotic sloughed epithelial cells and degenerating spermatozoa present in the lumens (Fig. 1e, upper panel). Abundant viral antigen staining was observed by IHC in the epididymal lumens (Fig. 1e, lower panel). In the epididymis at 14 d.p.i., the lining cells were low cuboidal with increased basophilia, reflecting loss of cells and attempted regeneration. The lumens contained numerous sloughed necrotic epithelial cells, degenerating spermatozoa and a small number of normal appearing spermatozoa (Fig. 1e, upper panel). Only scattered, infected lining epithelial cells were observed at day 14 but viral antigen was observed within the lumens affecting the sloughed cells and spermatozoa (Fig. 1e, lower panel). At 21 d.p.i, the epididymal lumens contained primarily necrotic debris and degenerated cells (Fig. 1e, upper panel). Abundant viral antigen staining was also observed within the epididymal lumens (Fig. 1e, lower panel).

**ZIKV infection results in a reduction in sperm parameters.** High levels of ZIKV RNA were detected in sperm isolated from the caudal epididymis at 7 d.p.i. and persisted until at least day 21 d.p.i. (Fig. 2a). Sperm cells from ZIKV-infected mice stained positive for ZIKV antigen at 21 d.p.i (Fig. 2b). At 7 d.p.i there was an increase in the percentage of fragmented sperm with dissociation of the sperm head from the tail in ZIKV-infected mice compared to uninfected mice (one-way ANOVA $P = 0.0473$), and at 14 d.p.i., the majority of sperm harvested from infected mice were fragmented (Fig. 2c). The mean total sperm counts in the ZIKV-infected mice at 21 d.p.i. were $29 \times 10^4$ ml$^{-1}$, thus significantly reduced compared to $48 \times 10^4$ ml$^{-1}$ in the uninfected mice (Mann–Whitney test $P = 0.0022$; Fig. 2d). Moreover, the percentage of sperm that were motile (Fig. 2e) and progressively motile (Fig. 2f) were significantly reduced in ZIKV-infected mice at 21 d.p.i. (one-way ANOVA $P = 0.0001$). In ZIKV-infected mice the percentage progressive motility became increasingly diminished as the infection progressed.

**ZIKV prME protects against ZIKV-induced testis damage.** In order to determine whether the vaccine could protect against ZIKV-induced damage to the male reproductive tract a ZIKV-prME DNA vaccination study was initiated. Five-week-old Ifnar1$^{-/-}$ mice received two vaccinations by the i.m. route with electroporation-mediated DNA delivery at 2-week intervals with

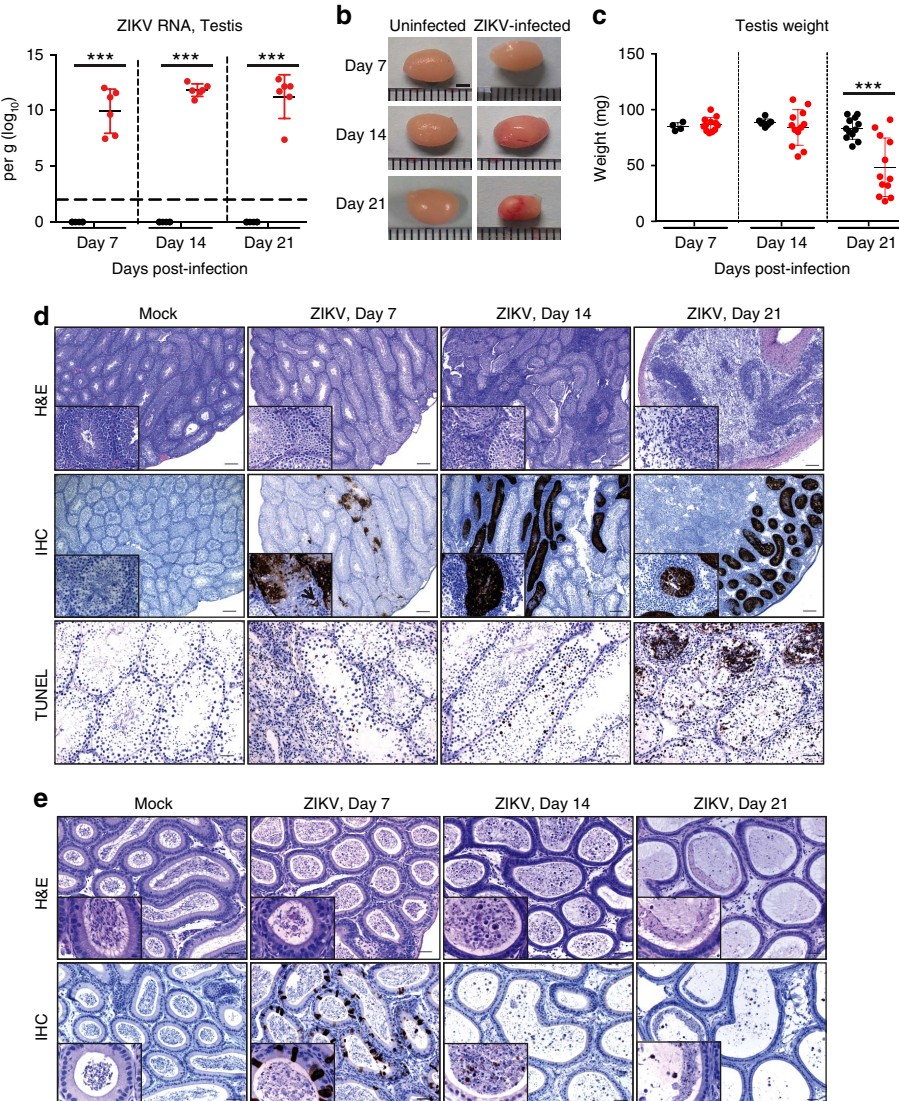

**Figure 1 | ZIKV infection of the testis and epididymis.** Ten- to eleven-week-old male Ifnar1$^{-/-}$ mice were inoculated with $5 \times 10^5$ p.f.u of ZIKV strain PRVABC59 by a subcutaneous route (s.c.) into each hind leg ($1 \times 10^6$ p.f.u. total) and compared to PBS mock-infected mice ($n = 6$). The testes and epididymides were collected at the indicated d.p.i. (**a**) ZIKV RNA copies in the testis of mice measured by quantitative reverse transcription (qRT)–PCR. Results are expressed as genome equivalents per gram of tissue. The horizontal hatched line indicates the limit of detection ($<100$ copies per ml). (**b**) Representative images of testis from mock-infected and ZIKV-infected mice at the indicated d.p.i. (**c**) Weight of the testis (two per mouse) from mock-infected and ZIKV-infected mice at the indicated d.p.i. (**d**) Histopathological analysis of the testis (upper panels), immunohistochemical labelling of ZIKV particles in the testis (middle panels), and TUNEL staining of testis (lower panels) at the indicated d.p.i. (**e**) Histopathological analysis of the epididymis (upper panels) and immunohistochemical labelling of ZIKV particles in the epididymis (lower panels) at the indicated d.p.i. Scale bars, 2 mm in **b** 200, 200, 50 μm (top, middle and lower panels) in **d** and 50 μm in **e**. The data shown are from one experiment that is representative of the same outcome in the two studies performed. Bars indicate mean values and error bars indicate s.d. Statistical differences are given (two-way ANOVA followed by the Bonferroni post-test). ***$P < 0.001$.

25 μg of the control vector (pVax1) or ZIKV-prME DNA vaccine (ZV-prME). Mice were challenged 2 weeks after the second immunization with a total of $1 \times 10^6$ p.f.u. of ZIKV strain PRVABC59 by the s.c. route. The testes and epididymis were harvested at 28 d.p.i. from sham-vaccinated and vaccinated mice and compared to testes obtained from age-matched phosphate-buffered saline (PBS) mock-infected and ZIKV-infected mice. Unvaccinated and sham-vaccinated mice that received a ZIKV challenge lost weight and 30% of the mice were euthanized due to weigh loss and other morbidity, including hind-limb paralysis (Supplementary Fig. 3a–c). Conversely, prME-vaccinated mice did not show signs of morbidity and mortality (Supplementary Fig. 3a–c), exhibiting normal pathology

comparable to unchallenged control mice. Following ZIKV challenge, high ZIKV RNA loads were detected in the testes of unvaccinated and sham-vaccinated mice, but no detectable virus was present in the testes of the synthetic prME-vaccinated mice (Fig. 3a). Moreover, there was an evident reduction in testis size (Fig. 3b) and weight (Fig. 3c) in all unvaccinated and sham-vaccinated mice that received a ZIKV-challenge compared to the uninfected and prME-vaccinated mice (two-way ANOVA $P = 0.0001$). The testis size (Fig. 3b) and weight (Fig. 3c) were not significantly different in sham-vaccinated compared to untreated-infected mice. Histopathological analysis revealed that unvaccinated or sham-vaccinated mice exposed to ZIKV suffered severe damage to the testis with the normal architecture disrupted

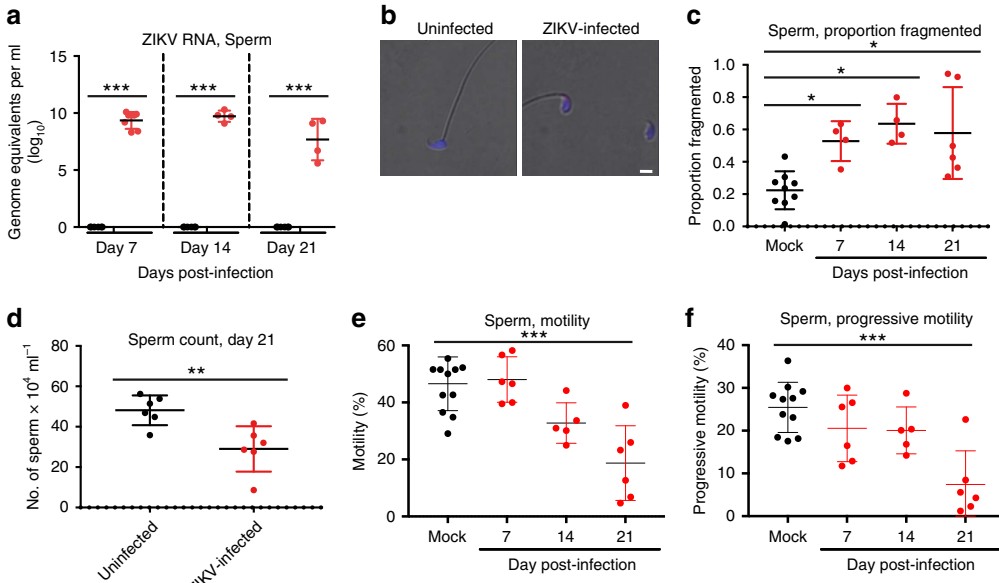

**Figure 2 | ZIKV infection of the sperm.** Ten- to eleven-week-old male Ifnar1$^{-/-}$ mice were inoculated with $5 \times 10^5$ p.f.u of ZIKV strain PRVABC59 by a subcutaneous route (s.c.) into each hind leg ($1 \times 10^6$ p.f.u. total) and compared to age-matched PBS mock-infected mice. Sperm form the caudal epididymis was collected immediately after euthanization. (**a**) ZIKV RNA copies in the caudal sperm of mice measured by qRT-PCR ($n = 4$ or greater). Results are expressed as genome equivalents per ml. The horizontal hatched line indicates the limit of detection (<100 copies per ml). (**b**) Immunofluorescence analysis of caudal sperm with anti-ZIKV antibody. Blue indicates DAPI staining and red represents ZIKV. (**c**) The proportion of fragmented sperm was assessed following staining with Diff-Quick solution. (**d–f**) Fertility parameters of caudal sperm were evaluated, including (**d**) concentration on day 21 after infection and (**e**) the percentage motility and (**f**) progressive motility at the indicated times post-infection. Scale bars, 2 μm in **b**. The data shown are from one experiment that is representative of the same outcome in the two studies performed. Bars indicate mean values and error bars indicate s.d. Statistical differences are given (**a**, two-way ANOVA followed by the Bonferroni post-test; (**c,e,f**) one-way ANOVA followed by Dunnett's test; **d**, Mann–Whitney test). *$P < 0.05$; **$P < 0.01$; ***$P < 0.001$.

with massive loss of seminiferous tubules and replacement by fibrous tissue and inflammatory cells, whereas prME-vaccinated mice showed normal phenotypes as for uninfected control mice (Fig. 3d and Supplementary Fig. 3d,e). The epididymal lumens of prME-vaccinated mice appeared unremarkable; however, the unvaccinated and sham-vaccinated mice contained primarily necrotic debris and degenerated cells (Fig. 3e). Viral antigen staining was observed within the epididymal lumens of unvaccinated and sham-vaccinated mice, but no staining was detected in prME-vaccinated mice (Fig. 3e). Immune cell infiltrates into the testes consisting of CD4$^+$, CD8$^+$ and NK cells (Supplementary Fig. 3d) and macrophages (Supplementary Fig. 3e) were observed in unvaccinated, but not prME-vaccinated mice exposed to ZIKV. At 77 d.p.i. the weight of the testes of unvaccinated and sham-vaccinated infected mice were significantly reduced compared to both mock-infected and prME-vaccinated infected mice, respectively (one-way ANOVA $P = 0.0001$) (Supplementary Fig. 4).

**ZIKV prME protects against ZIKV-induced damage to sperm.** High levels of ZIKV RNA were detected in sperm isolated from the caudal epididymis of unvaccinated and sham-vaccinated mice exposed to ZIKV, but signals from mock-infected and prME-vaccinated mice for these tissues were below the limit of detection of the assay (Fig. 4a). Sperm cells from unvaccinated and sham-vaccinated ZIKV-infected mice stained positive for ZIKV antigen at 28 d.p.i, although most cells were negative, whereas sperm cells from mock-infected and prME-vaccinated mice were all negative (Fig. 2b). We observed that the percentage of fragmented sperm with dissociation of the sperm head from the tail was significantly elevated in ZIVK-exposed unvaccinated and sham-vaccinated mice, compared to prME-vaccinated mice

(Fig. 4b,c; one-way ANOVA $P = 0.0464$). Moreover, the prME vaccine fully protected against the reduction in total sperm counts (Fig. 4d; one-way ANOVA $P = 0.0443$), motile sperm (Fig. 4e; one-way ANOVA $P = 0.0153$, Supplementary Movie 1), and progressively motile sperm (Fig. 4f; one-way ANOVA $P = 0.0048$). At 77 d.p.i. sperm parameters still had not returned to initial levels in unvaccinated and sham-vaccinated mice while the mock-infected and prME-vaccinated mice continued to have normal sperm parameters (Supplementary Fig. 4). There was no statistical difference in the level of ZIKV RNA in sperm or with any of the sperm parameters between the sham-vaccinated and infected mice.

**Discussion**
This work reports that immunization with a synthetic DNA vaccine encoding a ZIKV prME consensus sequence confers complete protection of mice against ZIKV-associated damage to the testis and prevents ZIKV persistence in the testis, epididymis and sperm. While the disease phenotype observed in ZIKV-infected Ifnar1$^{-/-}$ mice, including weight loss and neurological signs, is more severe than the mild illness typically associated with human infection; this model provides a stringent test to evaluate the ability of investigational therapeutics and vaccines to protect against ZIKV pathogenesis, including the mitigation and prevention of genitourinary signs and viral persistence within the male reproductive tract. Administration of the control plasmid vector, pVax1, before ZIKV infection resulted in an apparent reduction in ZIKV RNA detected in the testis (Fig. 3a), increase in testis weight (Fig. 3c), reduction in ZIKV RNA detected in the sperm (Fig. 4a) and improved sperm parameters (Fig. 4c–f) compared to non-treated ZIKV-infected mice. However, in all instances the differences in these

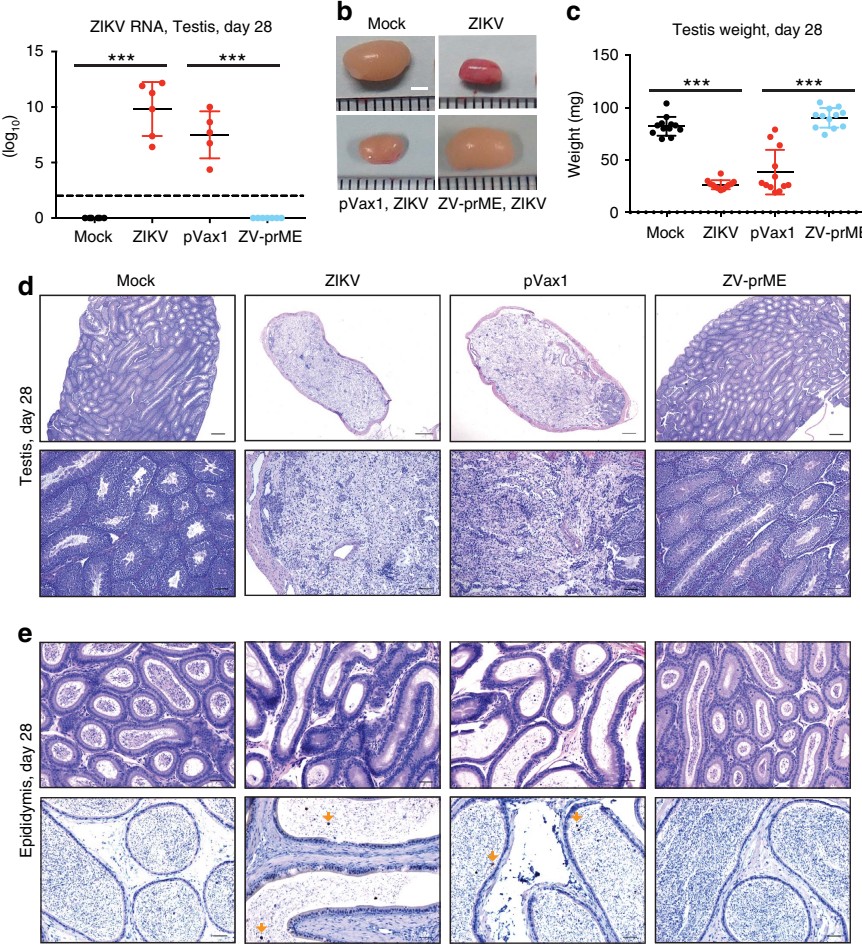

**Figure 3 | Protection from ZIKV-induced damage to the testis and epididymis by DNA immunization.** Five- to six-week-old male Ifnar1$^{-/-}$ mice received two vaccinations by the i.m. route with electroporation-mediated delivery at 2-week intervals with 25 μg of the control vector (pVax1) or ZIKV-prME DNA vaccine (ZV-prME). Mice were challenged 2 weeks after the second immunization with a total of $1 \times 10^6$ p.f.u. of ZIKV strain PRVABC59 by a subcutaneous route (s.c.) and compared to age-matched PBS mock-infected (mock) or ZIKV-infected mice (ZIKV). The testes and epididymides were collected at 28 d.p.i. (**a**) ZIKV RNA copies in the testis of mice measured by qRT-PCR ($n = 5$ or greater). Results are expressed as genome equivalents per gram of tissue. The horizontal hatched line indicates the limit of detection (<100 copies per ml). (**b**) Representative images of testis. (**c**) Weight of the testis (2 per mouse). (**d**) Histopathological analysis of the testis. (**e**) Immunohistochemical labelling of ZIKV particles in the epididymis. Arrows indicate cells positive for ZIKV antigen. Scale bars, 2 mm in (**b**), 200 μm in (**d**, top panel), 50 μm in (**d**, lower panel), 50 μm in (**e**, top panel), 10 μm in (**e**, lower panel). The data shown are from one experiment ($n = 6$) that is representative of the same outcome in the two studies performed. Bars indicate mean values and error bars indicate s.d. Statistical differences are given (two-way ANOVA followed by the Bonferroni post-test). ***$P = 0.001$.

parameters between pVax1 and non-treated infected mice were not statistically significant (ANOVA $P > 0.05$). We speculate that pVax1 treatment alone could stimulate a non-specific inflammatory response from the injection of naked DNA followed by electroporation. This phenomenon has been reported previously in vertebrates in part due to unmethylated CpG motifs within plasmid DNA molecules[27]. Future studies with larger group sizes could be performed to investigate this possibility.

ZIKV persistence in human semen[7–9] and sperm[10] has been documented for up to several months following the onset of symptoms, and pathological genitourinary symptoms have been described in infected men, including microhematospermia and dysuria[12,13,28]. However, deleterious effects on sperm parameters and fertility, transient or otherwise, in ZIKV-infected males, including those exposed to the virus *in utero*, have not been reported to this day. Studies on this subject will be critical to better understand the physiopathology of ZIKV in humans as well as the full utility of the mouse models of ZIKV infection. There is, however, precedent for such an occurrence: while typically spontaneously self-resolving, bilateral mumps orchitis

has been shown to result in suboptimal fertility in infected men[29]. Further, sexual transmission of ZIKV between humans has been well documented[7,11–14], and additional studies are likely warranted in a mouse model of sexual ZIKV transmission[30] to also assess the ability of investigational therapeutics and vaccines to prevent this from occurring.

Taken together, these data suggest that further studies are warranted to evaluate investigational therapeutics[31–34] and vaccines[20–24] as means to reduce ZIKV persistence in the male reproductive tract and to limit the potential for sexual transmission in humans.

## Methods

**Ethics statement.** The experiments described in this study were carried out at the National Microbiology Laboratory (NML) at the Public Health Agency of Canada as described in the Animal use documents No CSCHAH AUD# H-16-005 and and H-16-009 and were approved by the Animal Care Committee located at the Canadian Science Center for Human and Animal Health in accordance with the guidelines provided by the Canadian Council on Animal Care. All surgical procedures were performed under anesthesia induced and maintained with ketamine hydrochloride and xylazine, and all efforts were made to minimize

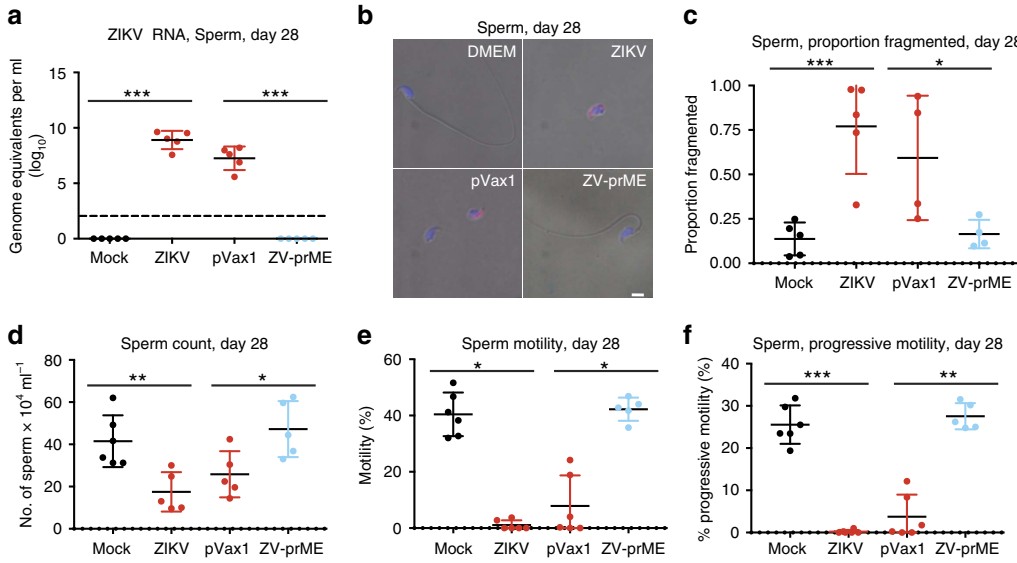

**Figure 4 | Protection from ZIKV-induced sperm damage by DNA immunization.** Five- to six-week-old male Ifnar1[−/−] mice received two vaccinations by the i.m. route with electroporation-mediated delivery at 2-week intervals with 25 μg of the control vector (pVax1) or ZIKV-prME DNA vaccine (ZV-prME). Mice were challenged 2 weeks after the second immunization with a total of $1 \times 10^6$ p.f.u. of ZIKV strain PRVABC59 by a subcutaneous route (s.c.) and compared to age-matched PBS mock-infected or ZIKV-infected mice. Sperm form the caudal epididymis was collected immediately after euthanization. (**a**) ZIKV RNA copies in the caudal sperm of mice measured by qRT-PCR ($n = 5$). Results are expressed as genome equivalents per ml. The horizontal hatched line indicates the limit of detection (<100 copies per ml). (**b**) Immunofluorescence analysis of caudal sperm with anti-ZIKV antibody. Blue indicates DAPI staining and red represents ZIKV. (**c**) The proportion of fragmented sperm was assessed following staining with Diff-Quick solution ($n = 4$). (**d–f**) Fertility parameters of caudal sperm were assessed ($n = 5$), including concentration (**d**), percentage motility (**e**) and progressive motility (**f**). Scale bars, 2 mm in **b**. The data shown are from one experiment ($n = 6$) that is representative of the same outcome in the two studies performed. Bars indicate mean values and error bars indicate s.d. Statistical differences are given (**a,c–f**, one-way ANOVA followed by Dunnett's test. *$P < 0.05$; **$P < 0.01$; ***$P < 0.001$.

animal suffering and to reduce the number of animals used. All infectious work was performed under biosafety level 2 (BSL-2) conditions.

**Animals and immunizations.** A ZIKV-prME consensus sequence was synthesized and cloned into the pVax1 expression vector (Genscript, NJ) under the control of the cytomegalovirus immediate-early promoter, as previously described[22]. Five- to six-week-old male mice lacking the type I interferon receptor (IFN-α/βR) (Ifnar1[−/−]) (B6.129S2-Ifnar1tm1Agt/Mmjax) were purchased from the Mutant Mouse Resource and Research Center (MMRRC) provided by The Jackson Laboratory. The mice were randomly assigned into their respective groups and housed and treated in a temperature-controlled, light-cycled facility in accordance with the Public Health Agency of Canada Institutional Animal Care Committee guidelines. Mice were immunized with 25 μg of DNA in a total volume of 30 μl of water delivered into the tibialis anterior muscle with *in vivo* EP delivery. *In vivo* EP was delivered with the CELLECTRA adaptive constant current EP device (Inovio Pharmaceuticals, PA) at the same site immediately following DNA injection. A three-pronged CELLECTRA minimally invasive device was inserted ∼2mm into the muscle. Square-wave pulses were delivered through a triangular 3-electrode array consisting of 26-gauge solid stainless steel electrodes and two constant current pulses of 0.1 Amps were delivered for 52 μs per pulse separated by a 1 s delay. Further protocols for the use of EP have been previously described in detail[35]. Mice were immunized two times at 2-week intervals and challenged 2 weeks after the final immunization. Blinding of the animal experiments were not performed, and samples sizes were not calculated *a priori*. No animals were excluded from the data analysis.

**Mouse challenge studies.** Ten- to eleven-week-old male Ifnar1[−/−] mice were challenged via the s.c. route with $5 \times 10^5$ p.f.u. in each leg in a volume of 50 μl (total dose of $1 \times 10^6$ p.f.u.) of ZIKV strain, ZIKV/Homo sapiens/PRI/PRVABC59/2015 (referred below as ZIKV, GenBank accession no. KX087101.2), provided by the Centers for Disease Control and Prevention (CDC). Post challenge, animals were weighed daily and body temperature was monitored with a s.c. temperature probe. In addition, animals were observed twice daily to assess any clinical signs of disease. At the indicated times after-infection mice were anesthetized with ketamine hydrochloride and xylazine and blood was removed from the organs by cardiac perfusion with 30 ml of ice-cold PBS prior to euthanasia. Tissues were harvested for downstream assays, as described below. Virus stocks were propagated in Vero cells (ATCC: CCL-81) that were confirmed to be free of mycoplasma and titrated by plaque assay, as described previously[22].

**Histology and IHC analysis.** Tissues were fixed in 10% neutral phosphate buffered formalin, routinely processed and sectioned at 5 μm. A set of slides were stained with hematoxylin and eosin for histopathologic examination. For IHC, the paraffin tissue sections were quenched for 10 min in aqueous 3% hydrogen peroxide and rinsed in MilliQ water. Epitopes were retrieved using Dako Target Retrieval solution (Dako, USA) in a Biocare Medical Decloaking Chamber. The staining was carried out using a Dako autostainer. Tissues were blocked with 10% normal goat serum in TBS buffer for 10 min when required. The primary antibody was an anti-FlaviVirus (D1-4G2-4-15) rabbit monoclonal antibody (Absolute Antibody Ltd., Oxford, UK) used at a dilution of 1:200 at room temperature for 30 min. They were then visualized using a horseradish peroxidase-labelled polymer, Envision + system (anti-rabbit) (Dako, USA) for 30 min and reacted with the chromogen diaminobenzidine (DAB). The sections were then counter stained with Gill's hematoxylin. Occurrence of apoptosis was detected using the TUNEL (terminal deoxynucleotide transferase-mediated dUTP nick-end labelling) technique. Five-micrometre paraffin-embedded tissue sections were run according to the peroxidase staining of paraffin-embedded tissue in the manual for the ApopTag Peroxidase *In Situ* Apoptosis Detection Kit (S7100) by Millipore. The sections were then counter stained with Gill's hematoxylin, dehydrated, cleared and cover slipped. For macrophage staining, anti-IBA1(Cat. #019-19741, Wako Pure Chemical Industries, Japan) primary antibody was used for 1 h at a dilution of 1:4,000. The GBI Labs Polink-2 HRP Plus Rabbit DAB detection System for IHC kit was used, according to manufacturer's instructions. DAB chromogen was applied for 2 min (Betazoid DAB from Biocare Medical) followed by Gill's III Hematoxylin for 1 min.

**Virus burden in tissues.** For infectious virus assays, a portion of the testis and epididymis were each harvested, weighed and placed in MEM supplemented with 2% heat-inactivated FBS, and then homogenized in a tissue homogenizer. Samples were spun down at 400g for 5 min and stored at −80 °C until later use. Testis homogenate was serially diluted 10-fold in MEM supplemented with 2% heat-inactivated FBS. One hundred microliter volumes of the dilutions were then added to 96-well plate in replicates of three of 95% confluent Vero cells and incubated for 1 h at 37 °C. The samples were removed and fresh medium was added and plates were scored for the presence of cytopathic effect on day 6 after infection. For qPCR assays testis were immersed in RNAlater (Ambion, Waltham, MA, USA) 4 °C for 1 week, then stored at −80 °C. The testis tissue was then weighed and homogenized in 600 μl RLT buffer using a TissueLyser (Qiagen, Valencia, CA, USA) with a stainless steel bead for 6 min at 30 cycles s$^{-1}$. A ZIKV-specific real-time RT–PCR assay was utilized for the detection of viral RNA[36]. RNA was reverse transcribed and amplified using the primers ZIKV 835

(5′-TTGGTCATGATACTGCTGATTGC-3′) and ZIKV 911c (5′-CCTTCCACAA
AGTCCCTATTGC-3′) and probe ZIKV 860FAM (5′-CGGCATACAGCATCAGG
TGCATAGGAG-3′) using the TaqMan Fast Virus 1-Step Master Mix (Applied
Biosystems, Foster City, CA, USA). A standard curve was generated in parallel for
each plate and used for the quantification of viral genome copy numbers. The
StepOnePlus Real-Time PCR System (Life Technologies Corporation, Carlsbad,
CA, USA) software version 2.3 was used to calculate the cycle threshold values, and
a cycle threshold value ≤38 for at least one of the replicates was considered
positive, as previously described.

**Flow cytometry and intracellular cytokine staining assay.** For intracellular
cytokines staining, single-cell suspensions of splenocytes were made by
homogenizing and processing the spleens through a cell strainer. Cells were then
re-suspended in ACK Lysing buffer (GibcoTM) for 1 min to lyse red blood cells
before two washes with PBS and final re-suspension in RPMI complete media
(RPMI 1640 + 10% FBS + 1% penicillin–streptomycin). Two million cells were
stimulated in 96-well plates with overlapping peptide pools spanning the entire
envelope protein of ZIKV for 6 h at 37 °C + 5% $CO_2$ in the presence of GolgiPlug
and GolgiStopTM (BS Biosciences). After 6 h, cells were collected and stained in
FACS buffer with a panel of surface antibodies containing Fixable Viability Stain
780, PerCP-Cy5.5 anti-CD3 (1:400), Alexa Fluor 700 anti-CD45 (1:400), BV786
anti-CD4 (1:400) and BV421 anti-CD8 (1:200) (BD Biosciences, Cat. #551163,
#560510, #563331, #563898) for 30 min at 4 °C. Cells were washed three times
before being fixed with Cytofix/Cytoperm (BD Biosciences) for 30 min at 4 °C.
Cells were washed with Perm/Wash buffer (BD Biosciences) three times before
intracellular staining with BV650 anti-IFNγ (1:100), Alex Fluor 488 anti-IL-2
(1:200), PE-Cy7 anti-TNFα (1:400) (BD Biosciences, Cat. #563854, 557725,
557644) and Alexa Fluor 647 anti-human/mouse Granzyme B (1:100) (BioLegend)
for 30 min at 4 °C. Cells were then washed three times with Perm/Wash buffer
before suspension in FACS buffer and acquisition on a BD LSRII. All results were
analysed using FlowJoTM v.10.0 (TreeStar).

**Immunofluorescence analysis.** Mature sperm were deposited on poly-D-lysine-
coated cover slips. Cells were fixed with 4% paraformaldehyde/sucrose for 10 min
at room temperature. Cells were permeabilized with 0.5% Triton X-100
(Sigma-Aldrich) in PBS for 10 min. Samples were then incubated with rabbit
anti-Flavivirus group antigen primary antibody, clone D1-4G2-4-15 (1:20)
(EMD Millipore) for 2 h at room temperature, washed with 0.1% Triton X-100
and PBS, followed by incubation with goat anti-rabbit secondary conjugated to
Alexa-Fluor 647 (1:200; Life Technologies) for 1 h at room temperature, washed
once with 0.1% Triton X-100 and PBS and mounted using ProLong Gold
(Life Technologies). Images were acquired using a Zeiss LSM 700 confocal
microscope.

**Sperm analysis.** Mature sperm was collected from the caudal epididymis. In brief,
one of the epididymides were randomly selected for collection from euthanized
mice and placed in a 35-mm petri dish containing 1 ml of PBS. The epididymides
were dissected using a dissecting microscope and 18G needles; sperm was allowed
to swim out for 30 min at 37 °C. Total counts and motility were evaluated within
60 min of collection, as previously described in the WHO laboratory manual for the
Examination and processing of human semen[37]. Sperm RNA was isolated using
Qiazol Lysis Reagent (Qiagen) and choloform extraction followed by purification
with the QIAamp Viral RNA Mini kit (Qiagen).

**Statistical analysis.** Results were analysed and graphed using Prism 7 software
(Graphpad Software). As appropriate, statistical analyses were performed using
two-way ANOVA with Bonferroni's post-test, one-way ANOVA with Dunnett's
test or the Mann–Whitney test.

**Data availability.** All relevant data are available from the authors upon request.

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

# Acknowledgements

We thank Michelle French and Kimberly Azaransky of the Veterinary Technical Services
at the National Microbiology Laboratory (NML) for their technical assistance during
the course of this work and Estella Moffat, Brad Collignon and Jill Graham of NCFAD
for pathology technical work. We thank the Centers for Disease Control and Prevention
(CDC) for providing the ZIKV isolate. We thank Sonja M. Best of the National
Institute of Allergy and Infectious Disease for consultations regarding the ZIKV IHC. All
protocols and procedures were approved by the institutional research ethics committees
of the Canadian Science Centre for Human and Animal Health (CSCHAH) and Laval
University. This study was supported by the Public Health Agency of Canada and D.B.W.
was supported in part by the Intramural Research Program, National Institute of
Allergy and Infectious Diseases, National Institutes of Health (grant R01-AI092843);
DARPA- PROTECT to D.B.W. and K.M.; BMGF to D.B.W. and K.M. and Inovio
Pharmaceuticals to D.B.W. and K.M.

## Author contributions

B.D.G., K.M., B.M.W., A.M., M.H., J.A., D.R.S., C.R., T.R., M.-A.D.L.V., J.P., S.K., K.N.T., K.L.F., C.D.G., G.S., L.S. performed the studies and analysed the data. J.S., G.M., V.S., Y.K.P., J.N.M., N.Y.S., J.J.K., X.-J.Y., A.B., R.L., G.B., S.A.B. provided scientific support as well as assistance in discussing and interpreting experimental results as well as related assays. B.D.G., K.M., D.K., C.E.-H., D.S., D.B.W., G.P.K. designed and supervised the experiments and data generation in addition to writing the manuscript. All the authors have read and commented on the final manuscript and have agreed to its submission.

## Additional information

**Competing interests:** D.B.W. has grant funding, participates in industry collaborations, has received speaking honoraria and fees for consulting. This service includes serving on scientific review committees and advisory boards. Remuneration includes direct payments and/or stock or stock options and in the interest of disclosure; therefore, he notes potential conflicts associated with this work with Inovio where he serves on the BOD, Merck, VGXI, OncoSec, Roche, Aldevron and possibly others. The remaining authors declare no conflict of interest.

**Publisher's note**: 

