## [Peer Review File · Nature Communications]

Reviewers' comments:

Reviewer #1 (Remarks to the Author):

As the authors point out, there are a large number of Zika vaccine candidates in development. Some of the preclinical studies have been published, including one by the authors, and some have advanced to clinical evaluation. While it is clear that candidate vaccines can eliminate Zika virus (ZIKV) viremia, weight loss and clinical signs in animal models, there are still major questions regarding the ability of vaccines to induce sterilizing immunity or prevent persistence of virus in privileged sites such as the reproductive tissues, including the testes.

This manuscript investigates the ability of a candidate Zika DNA vaccine to protect against infection of the testes in a mouse model. Expanding on their previously published protection of interferon alpha/beta receptor knockout mice against ZIKV challenge, the authors demonstrate the vaccine protect mice against damage to the testes and eliminates virus persistence in the testes and sperm. This is an important study as it demonstrates a proof of concept that a ZIKV vaccine can protect virus infection of reproductive tissues in a sensitive mouse model.

The authors provide a detailed analysis of the male reproductive tissues in ZIKV infected mice followed by the effects/protection of vaccination, and present very interesting data on the ability of the candidate vaccine to protect against infection of the testes and sperm.

Line 162-164/Figures 3a, 3c, 4a, 4c-f. It looks like the plasmid vector alone has some benefits in terms of reduction in weight of testes and viral RNA in testes/sperm. Are the weights with little reduced weight and reduced viral load matched pairs of testes from the same animals, i.e., some animals react different to others? How do the authors explain the apparent benefits of the plasmid vector? Have the authors looked for the presence of the plasmid in the male reproductive tissues by PCR? There appears to be no statistical analysis on the benefits of the plasmid vector only vs virus infection. This would be helpful, although as said above I wonder if pairs of testes should similar data.

Have the authors investigated how long protective immunity lasts and whether or not there is infection of the testes at later times post infection.

It is stated the data shown is representative of one of two experiments. The authors need to state whether or not the two studies gave the same results.

The interferon alpha/beta/gamma receptor knockout mouse model is more sensitive (i.e., ZIKV is uniformly lethal) than the current model in this paper. Have the authors looked at

this model with this vaccine candidate in a similar way as in this paper?

Reviewer #2 (Remarks to the Author):

A host of emerging evidence has demonstrated that Zika virus infection negatively impacts testicular function in mice, in particular has a deleterious effect on spermatogenesis. Effective strategies to prevent or limit testicular damage due to Zika infection have not been devised, thus the present study address an area of great significance. Overall, the study is really well done and the data are convincing. This is one of the rare occasions that I do not have major concerns with the experimental approaches or interpretations of the findings. However, there are a couple of minor issues that the authors may want to take into account.

Although the evidence provided in Figure 1 does support a claim of damage to the testis following Zika infection, evidence of spermatogonial and spermatocyte degeneration is not compelling. Clearly, degeneration of the seminiferous epithelium is occurring but direct evidence for an impact on spermatogonia and spermatocytes as indicated in lines 107-109 is not provided. I suggest that the authors soften this statement by indicating that, "damage to the seminiferous epithelium was evident which may include degeneration of the spermatogenic lineage", rather than single out spermatogonia and spermatocytes.

The IHC analyses presented for Figure 1d do not allow for clearly discerning viral antigen in spermatogonia, spermatocytes, and Sertoli cells. Again, my suggestion is to soften the statement by indicating that, "viral antigen was present in the seminiferous epithelium which includes all layers of maturing spermatogenic cells and Sertoli cells", rather than single out specific cell types.

Reviewer #3 (Remarks to the Author):

This manuscript describes the application of a previously tested DNA vaccination against Zika virus (ZIKV) in a model of virus persistence in males. The vaccine encodes ZIKV PrME, and when administered to male ifnar -/- mice beginning at 5 weeks of age, conferred complete protection against ZIKV persistence in the testes and sperm, and prevented ZIKV-induced testicular and sperm pathology. These data complement previous data demonstrating that the vaccine is also protective against systemic disease in this animal model, but fails to demonstrate reduced transmission. Concerns about redundancy with previous published data as well as the value of the extreme testicular pathology in ifnar-/- for human disease dampens enthusiasm.

Major Critiques:

- Unfortunately, another group already published that ZIKV induces extreme damage to the

testes of ifnar^{-/-} male mice, resulting in reduced testosterone, dramatic pathology, and reduced sperm counts and motility (Govero et al. 2016 Nature); thus, the data presented in Figures 1-3 of the current manuscript merely confirm a phenotype already reported. It is only the vaccine data in Figure 4 that are novel, which does not seem sufficient for publication on its own. It is surprising that the authors never test whether ZIKV infection of ifnar^{-/-} and the presence of virus in the testes results in transmission of virus to females during mating. Further, it is imperative that they show that receipt of the ZIKV PrME vaccine reduces sexual transmission. Without demonstrating sexual transmission in this model, the value of this model is questionable because to date there are no clinical data suggesting that men infected with ZIKV are infertile or present with extreme testicular pathology as is observed in ifnar^{-/-} mice. Thus, what is this model system really modeling in humans?

- The data presented in Figures 1 and 2, characterizing the pathogenesis of ZIKV in the testes of ifnar^{-/-} mice (in the absence of vaccination) compares tissue viral burden, testicular size and pathology, and sperm count and motility across 3 time points (7, 14, and 21 days post-infection). The same outcome measures are characterized for animals following vaccination/challenge, but only at 28 days post-infection. What does viral load, testicular size, and sperm count/motility look like at earlier time points (e.g. 7, 14, and 21 days post-challenge)? Does the vaccine protect against infection of testes and sperm altogether? Or does early infection take place, and productive replication and persistence in the testes is inhibited?

- Figures 1 and 2 provide very detailed characterization of ZIKV-induced testicular and sperm pathology, with the phenotype at 21 days post-infection appearing quite severe (i.e. 50% reduction in testicular weight and significant disruption of normal testicular architecture). Have similar outcomes been observed in human males infected with ZIKV? Have there been reports of testicular atrophy or reduced sperm counts/motility in men who have tested positive for the virus in semen? It may be that the ifnar^{-/-} demonstrates an exaggerated phenotype (similar to systemic disease), which warrants discussion. Could the lack of IFN signaling be contributing to this extreme phenotype in their mouse model?

Reviewers' comments:

We thank the reviewers very much for the constructive comments. The manuscript has importantly benefited from the integration of all the suggestions and comments, and we hope that it will now be satisfactory for publication in *Nature Communications*.

Reviewer #1 (Remarks to the Author):

As the authors point out, there are a large number of Zika vaccine candidates in development. Some of the preclinical studies have been published, including one by the authors, and some have advanced to clinical evaluation. While it is clear that candidate vaccines can eliminate Zika virus (ZIKV) viremia, weight loss and clinical signs in animal models, there are still major questions regarding the ability of vaccines to induce sterilizing immunity or prevent persistence of virus in privileged sites such as the reproductive tissues, including the testes.

This manuscript investigates the ability of a candidate Zika DNA vaccine to protect against infection of the testes in a mouse model. Expanding on their previously published protection of interferon alpha/beta receptor knockout mice against ZIKV challenge, the authors demonstrate the vaccine protect mice against damage to the testes and eliminates virus persistence in the testes and sperm. This is an important study as it demonstrates a proof of concept that a ZIKV vaccine can protect virus infection of reproductive tissues in a sensitive mouse model.

The authors provide a detailed analysis of the male reproductive tissues in ZIKV infected mice followed by the effects/protection of vaccination, and present very interesting data on the ability of the candidate vaccine to protect against infection of the testes and sperm.

Line 162-164/Figures 3a, 3c, 4a, 4c-f. It looks like the plasmid vector alone has some benefits in terms of reduction in weight of testes and viral RNA in testes/sperm.

We agree with the reviewer that there is an apparent tendency for reduced weight loss and lower viral burden in the testis tissue and sperm in the mice administered the control plasmid vector, pVax1, prior to infection compared to mice that were only infected. Further, the negative effects of viral infection on the sperm parameters appeared to be less severe in the mice administered pVax1 prior to infection, compared to infected mice. In all these instances, however, statistical analysis revealed that the apparent differences were not significant (please see the Table below for a summary). We have added the following sentence to the Results section to clarify this point:

Lines 177-178 and lines 205-206, respectively:

“The testis size (Fig. 3b) and weight (Fig. 3c) were not significantly different in sham vaccinated compared to infected mice “

“There was no statistical difference in the level of ZIKV RNA in sperm or with any of the sperm parameters between the sham-vaccinated and infected mice. “

Figure	Panel	Data	Statistical test	Significance (pVax1 versus ZIKV)
3	A	qPCR testes	2-way ANOVA	no (P = 0.0637)
3	C	testes weight	2-way ANOVA (Y=log(Y) transform)	no (P = 0.0555)
4	A	qPCR sperm	1-way ANOVA	no (P = 0.1787)
4	C	% fragmented	1-way ANOVA	no (P = 0.7614)
4	D	count	1-way ANOVA	no (P = 0.2122)
4	E	motility	1-way ANOVA	no (P = 0.9310)
4	F	prog motility	1-way ANOVA	no (P = 0.1382)

Are the weights with little reduced weight and reduced viral load matched pairs of testes from the same animals, i.e., some animals react different to others?

This is an interesting question, and we think it is likely that host factors are playing a role in susceptibility to ZIKV infection. The data presented represent the weights and viral load from the same animals; however, the testes chosen for virus quantification versus pathology were chosen at random. While the testicular atrophy was bilateral it was often more severe on one side than the other. Viral load did not correlate with decreased testis size, which is very interesting, but this may be due to the fact that the viral loads were consistently very high. A study specifically designed to address this point would be required.

How do the authors explain the apparent benefits of the plasmid vector? Have the authors looked for the presence of the plasmid in the male reproductive tissues by PCR?

We speculate that the innate immune response may be activated by the plasmid, as has been described in the past with CpG motifs, and there is a possibility that the vectors contain cryptic promoter sequences that could generate unexpected transcripts and proteins that are immunogenic or immunostimulatory in some way, but that we would need larger group sizes (estimated n = 27) to detect this potential difference. We also cannot rule out variation in our infection model. In our studies, we have found that shipments of *infa1-/-* mice on a C57BL/6 background that are challenged beyond 6 weeks of age display apparent variability in susceptibility to lethal infection and viral burden in the tissues when administered the same titer of virus from aliquots of the same batch of virus stocks. This difference to date; however, has also not been statistically significant. We have gone back and determined the exact age reported for the mice, but these small differences in age did not correlate with susceptibility to infection.

This is an interesting point about the plasmid DNA that has been raised by the reviewer that we have also considered. In previous work related to FDA licensure of other similar DNA vaccine constructs, we have demonstrated that the vaccine DNA is only found with any frequency locally at the site of injection. A biodistribution study was conducted to examine the distribution and potential integration of an influenza cocktail vaccine (INO-3510) that uses the same plasmid vector (pVax1) used in the current study when administered as a single ID dose followed by EP using the CELLECTRA®-3P device to NZW rabbits over a 90-day period.

The plasmids generally remained localized in the skin/subcutis and muscle at the injection sites on study day (SD) 9. Plasmid clearance by SD 30 was nearly complete in the muscle from Group 1 (ID injection control, i.e. no EP) and Group 2 animals (ID injection with EP) and from the skin/subcutis of Group 2 animals. Two Group 1 animals (out of 10) were noted to have >30,000 copies (34,836 and 74,800 copies) of plasmid in samples from the skin/subcutis on SD 90. Furthermore, the clearance of plasmid from the skin/subcutis of Group 1 animals on SD 90 was >99.9998%, when compared to estimated dose injected at SD 0. All Group 2 animals, (ID followed by EP), showed lack of persistence with marked plasmid clearance as judged by <30,000 copies in all skin/subcutis and muscle samples at SD 90 with a significant decay in plasmid copies from SD 0 to SD 9, and further to SD 90. These data were not included because they are part of the large toxicology profile submitted to the FDA for clinical trials (data owned by Inovio), however, we thought they should be shared here to answer the question hoping that it is helpful.

There appears to be no statistical analysis on the benefits of the plasmid vector only vs virus infection.

Thank you for identifying this shortcoming. Following the suggestion of the reviewer we have revised the text, as described above.

This would be helpful, although as said above I wonder if pairs of testes should similar data.

Interestingly, viral load was found to not correlate with decreased testis size, and the damage to the testis was often asymmetrical. The testis from each animal; however, were randomly assigned for either virological or pathology analyses, so we do not have data to directly compare the testis from a given animal.

Have the authors investigated how long protective immunity lasts and whether or not there is infection of the testes at later times post infection.

This is an interesting point. We have determined that immunity lasts for many months at a minimum. Our previous study showed that antibody titers had peaked by the 35 days after immunisation and had not declined as of day 60 after vaccination (Fig. 3, Muthumani et al., 2016 ,NPJ Vaccines). Unfortunately, a limitation of all the current murine models of ZIKV infection is that the mice are uniformly resistant to lethal infection beyond week 11 of age, rendering it impossible to directly test the protection conferred upon challenge at several months post-vaccination.

It is stated the data shown is representative of one of two experiments. The authors need to state whether or not the two studies gave the same results.

We thank the reviewer for this observation. The text was modified as advised to reflect the same outcome in both biological replicates.

Lines 511-512, 535-536, 554-555, 585-586

“The data shown are from one experiment that is representative of the same outcome in the two studies performed”

The interferon alpha/beta/gamma receptor knockout mouse model is more sensitive (i.e., ZIKV is uniformly lethal) than the current model in this paper. Have the authors looked at this model with this vaccine candidate in a similar way as in this paper?

This is an interesting point. To our knowledge while all the data reported for ZIKV-challenged IFN- $\alpha/\beta/\gamma$ -R-/- (A129 background) shows uniform lethality in response to ZIKV infection, the studies reporting the AG129 models used mice at 4 weeks of age (Lazear et al., 2016, Cell Host & Microbe), 3 weeks of age (Rossi et al., 2016 Am. J. Trop. Med. Hy), and 8 weeks of age (Aliota et al., 2016 PLoS NTD). We agree with the review that it remains a distinct possibility that AG-129 would be susceptible at ages greater than 10 weeks of age (as required for our vaccine studies). The IFN- α/β -R-/- mice used in our studies are also uniformly susceptible to lethal infection up to 6 weeks of age at the time of challenge. We made sure to obtain mice at 5-6 weeks of age, so that they could be as young as possible following the prime and boost at the time of challenge. During pilot studies when we were initially developing this model we tested IFN- $\alpha/\beta/\gamma$ -R-/- mice for susceptibility and found the lethality to be uniform, but delayed compared to IFN- α/β -R-/- mice. Additional studies to deplete IFN γ to see if depletion had a deleterious effect were inconclusive (unpublished data). Further, to our knowledge these mice are only commercially available from BK Universal in the United Kingdom. In future studies, it would be interesting to test whether testicular atrophy occurs in these mice since IFN γ expression could influence activation status of T cells that have infiltrated into the testis tissue.

Reviewer #2 (Remarks to the Author):

A host of emerging evidence has demonstrated that Zika virus infection negatively impacts testicular function in mice, in particular has a deleterious effect on spermatogenesis. Effective strategies to prevent or limit testicular damage due to Zika infection have not been devised, thus the present study address an area of great significance. Overall, the study is really well done and the data are convincing. This is one of the rare occasions that I do not have major concerns with the experimental approaches or interpretations of the findings. However, there are a couple of minor issues that the

authors may want to take into account.

Although the evidence provided in Figure 1 does support a claim of damage to the testis following Zika infection, evidence of spermatogonial and spermatocyte degeneration is not compelling. Clearly, degeneration of the seminiferous epithelium is occurring but direct evidence for an impact on spermatogonia and spermatocytes as indicated in lines 107-109 is not provided. I suggest that the authors soften this statement by indicating that, “damage to the seminiferous epithelium was evident which may include degeneration of the spermatogenic lineage”, rather than single out spermatogonia and spermatocytes.

Thank you, agreed. Following the suggestion of the reviewer we have softened the statement to read,

“Microscopic evidence of damage to the seminiferous epithelium was evident at 7 d.p.i. which may include degeneration of the spermatogenic lineage (Fig. 1d, upper panel).”

Lines 119-120

The IHC analyses presented for Figure 1d do not allow for clearly discerning viral antigen in spermatogonia, spermatocytes, and Sertoli cells. Again, my suggestion is to soften the statement by indicating that, “viral antigen was present in the seminiferous epithelium which includes all layers of maturing spermatogenic cells and Sertoli cells”, rather than single out specific cell types.

Thank you. Following the suggestion of the reviewer we have softened the statement to read,

Line 121-123

“Viral antigen was observed by immunohistochemistry (IHC) multifocally throughout the seminiferous epithelium which includes all layers of maturing spermatogenic cells and Sertoli cells (Fig. 1d, middle panel)”

Reviewer #3 (Remarks to the Author):

This manuscript describes the application of a previously tested DNA vaccination against Zika virus (ZIKV) in a model of virus persistence in males. The vaccine encodes ZIKV PrME, and when administered to male ifnar -/- mice beginning at 5 weeks of age, conferred complete protection against ZIKV persistence in the testes and sperm, and prevented ZIKV-induced testicular and sperm pathology. These data complement previous data demonstrating that the vaccine is also protective against systemic disease in this animal model, but fails to demonstrate reduced transmission. Concerns about redundancy with previous published data as well as the value of the extreme testicular pathology in ifnar-/- for human disease dampens enthusiasm.

Major Critiques:

- Unfortunately, another group already published that ZIKV induces extreme damage to the testes of ifnar^{-/-} male mice, resulting in reduced testosterone, dramatic pathology, and reduced sperm counts and motility (Govero et al. 2016 Nature); thus, the data presented in Figures 1-3 of the current manuscript merely confirm a phenotype already reported. It is only the vaccine data in Figure 4 that are novel, which does not seem sufficient for publication on its own.

We agree with the reviewer that our data (in Figures 1 and 2) is consistent with the data reported in Nature (Govero et al. 2016). A notable difference; however, is that our model employed IFN- α/β -R^{-/-} mice, whereas the other study used transient depletion of the IFN- α/β receptor using anti-IFN- α/β antibodies, and our studies employed a contemporary isolate of ZIKV from Puerto Rico, whereas the other work employed a mouse-adapted African strain of ZIKV. Our work was well under way prior to the publication of the other manuscript, and we made the decision to use ifnar¹-/- mice since we viewed it as a possibly more consistent challenge model for testing prophylactic and therapeutic options. That being said we never did a side-by-side comparison between the 2 models. For this reason, we estimated that this important distinction required us to demonstrate the robustness of the model we were using prior to showing the effectiveness of the vaccine for protection against damage to the testis and sperm. It is also possible that the challenge model we used here may provide a higher benchmark for protection (considering the timeline) than transiently induced depletions in wildtype mice (that are naturally resistant). It is perhaps for this reason that we observed a greater degree of fragmentation to sperm cells than the previously published study. Figure 3 and 4; however, do not overlap with the prior study since it reports the effects of prior vaccination.

It is surprising that the authors never test whether ZIKV infection of ifnar^{-/-} and the presence of virus in the testes results in transmission of virus to females during mating. Further, it is imperative that they show that receipt of the ZIKV PrME vaccine reduces sexual transmission. Without demonstrating sexual transmission in this model, the value of this model is questionable because to date there are no clinical data suggesting that men infected with ZIKV are infertile or present with extreme testicular pathology as is observed in ifnar^{-/-} mice. Thus, what is this model system really modeling in humans?

These are very good points. We have initiated studies to evaluate fertility changes in mice in an ongoing project as well as to evaluate male-to-female transmission. Thus far, we see no transmission from ifnar¹-/- to wildtype mice and studies to evaluate transmission to ifnar¹-/- remain to be done. In early fertility studies that we are conducting we have observed that even microscopic ZIKV-induced damage to the testis appears to result in a decrease in fertility. Just after receiving the reviewer's comments, a sexual transmission model was reported that employed AG-129 mice (Tang et al., 2017 Cell Reports), and future studies could employ this model to address this research question.

When we noted this phenomenon to the testis in July 2016, [redacted]. Lastly, we would like to note that hematospermia (blood in the semen) has been identified in infected men, so tissue damage is possible in men as with this instance where a patient that had fully recovered from the infection then noted blood in the semen several weeks after infection (Musso et al., 2015 EID). Further, in our NHP model of ZIKV infection we have identified persistent foci of ZIKV antigen in the testes of immunocompetent NHPs (Osuna et al., 2016 Nature Med). We also suggest that developing fetuses may be susceptible to damage to the testis, and we have contacted collaborators to investigate this troubling possibility. Lastly, measles infection has been known to cause fertility issues in men that are infected within a certain age, and it remains to be seen if a similar situation is occurring in humans upon infection with ZIKV. The results of all these ongoing studies will be of the utmost importance. For these reasons, we believe that showing without delay that vaccination can prevent damages to the testes in an animal model is of importance also to inform ongoing clinical trials/studies evaluating vaccine candidates (similarly as it was critical to communicate the damages observed in Zika infected mice before this was confirmed or not in humans).

- The data presented in Figures 1 and 2, characterizing the pathogenesis of ZIKV in the testes of ifnar -/- mice (in the absence of vaccination) compares tissue viral burden, testicular size and pathology, and sperm count and motility across 3 time points (7, 14, and 21 days post-infection). The same outcome measures are characterized for animals following vaccination/challenge, but only at 28 days post-infection. What does viral load, testicular size, and sperm count/motility look like at earlier time points (e.g. 7, 14, and 21 days post-challenge)? Does the vaccine protect against infection of testes and sperm altogether? Or does early infection take place, and productive replication and persistence in the testes is inhibited?

We agree with the reviewer that it would be interesting to examine the testis and epididymis tissues at earlier times post-infection. In our analyses, we have not detected any evidence of prior infection of the testis, epididymis or sperm in any vaccinated mice that have been challenged with ZIKV. We cannot rule out prior infection of the testis and the possibility that the amount of virus present be below the level of detection of the assay. We expect, however, that since the testis are an immune privileged organ and that the mice lack the IFN- α/β receptor that we would be able to detect viral RNA if the virus had gained entry into the testis since it would be able to replicate unchecked. Further, in vaccinated mice we did not observe any immune infiltration into the seminiferous tubules; whereas, there were a great deal of immune infiltrates that had crossed the blood-testes barrier (BTB) in the unvaccinated mice. Lastly, in all repetitions of infection studies performed thus far, viral RNA was found to persist in macrophages within the seminiferous tubules until at least day 28 post-infection, and we expect that we would be able to detect viral RNA in macrophages if the BTB had been compromised. Lastly, there is evidence that any disruption of the BTB (even physical damage) will expose sperm germ cell antigen to the immune system and result in some degree of autoimmune orchitis. No reduction in size or weight of the testes was observed in prior vaccinated mice.

- Figures 1 and 2 provide very detailed characterization of ZIKV-induced testicular and sperm pathology, with the phenotype at 21 days post-infection appearing quite severe (i.e. 50% reduction in testicular weight and significant disruption of normal testicular architecture). Have similar outcomes been observed in human males infected with ZIKV? Have there been reports of testicular atrophy or reduced sperm counts/motility in men who have tested positive for the virus in semen? It may be that the ifnar -/- demonstrates an exaggerated phenotype (similar to systemic disease), which warrants discussion. Could the lack of IFN signaling be contributing to this extreme phenotype in their mouse model?

For human data please see the response just above to the point raised by this reviewer. For the time point, we selected day 28 after infection for our vaccine analyses since the tissue destruction and sperm damage was most pronounced at that time point, which increased the statistical power of our study to detect a significant benefit. We have added the following lines to the Discussion to expand on this point.

Lines 218-221. “The use of ifnar1^{-/-} mice with impaired innate immune signaling can be expected to exaggerate the disease phenotype, and we therefore suggest that this model provides a robust measure of vaccine-induced protection of the male reproductive system.”

REVIEWERS' COMMENTS:

Reviewer #1 (Remarks to the Author):

The author's revised manuscript is an improvement of the original version and they have addressed the points raised by all here reviewers well.

My only criticism is that the comments in the rebuttal to my (Reviewer #1) comments about the apparent benefit of the vector only have not made it to the revised paper. I believe that these points will interest readers and should be included in the revised paper.

Reviewer #3 (Remarks to the Author):

None of our concerns from the initial review were addressed in the revised manuscript.

1. The authors should acknowledge the previous published study that already demonstrated that in the absence of type I IFN signaling, Zika virus (ZIKV) can replicate in the testes and is found in the sperm. The distinctions between Govero et al. 2016 and the current study are minor differences that should be acknowledged in the introduction and not just in the response to the reviewers.

2. Because the data in Figures 1 and 2 are redundant of previously published work (see Govero et al. 2016), this paper should be focused on the value of the vaccine candidate. Showing that the vaccine reduces mortality and infection of reproductive tissues in males is a start. But showing that the vaccine eliminates persistent infection (Reviewer 1) or transmission of virus to females (Reviewer 3) is necessary. Both requests were dismissed and without these additional, rigorous experiments, the data presented in the current manuscript are preliminary.

3. The relevance of extreme testicular pathology in the absence of type I IFN signaling in mice to the human condition should not just be addressed in the response to the reviewer, but should be detailed in the manuscript. What is provided as a response seems to be hearsay if nothing has been published documenting testicular pathology in humans, to date. There are no published data indicating that ZIKV infection makes males sterile, or there would be no sexual transmission. For this to be a relevant observation for vaccines or other therapeutics, there needs to be relevance to the human condition and this is not addressed, even as a limitation of the model, in the manuscript.

Reviewers' comments:

We thank the reviewers very much for their additional feedback. The manuscript has benefited from the incorporation of the suggestions and comments. We hope that the manuscript has been suitably revised for publication in *Nature Communications*.

Reviewer #1 (Remarks to the Author):

The author's revised manuscript is an improvement of the original version and they have addressed the points raised by all here reviewers well.

My only criticism is that the comments in the rebuttal to my (Reviewer #1) comments about the apparent benefit of the vector only have not made it to the revised paper. I believe that these points will interest readers and should be included in the revised paper.

Thank you, agreed. Following the suggestion of the reviewer we have included the following comments related of the pVax1 vector and the apparent benefit of administration of the vector only prior to infection in the manuscript.

Lines 223-233:

“Administration of the control plasmid vector, pVax1, prior to ZIKV infection resulted in an apparent reduction in ZIKV RNA detected in the testis (Fig. 3a), increase in testis weight (Fig 3c), reduction in ZIKV RNA detected in the sperm (Fig. 4a), and improved sperm parameters (Fig. 4c-f) compared to non-treated ZIKV-infected mice. However, in all instances the differences in these parameters between pVax1 and non-treated infected mice were not statistically significant (ANOVA $p > 0.05$). We speculate that pVax1 treatment alone could stimulate a non-specific inflammatory response from the injection of naked DNA followed by electroporation. This phenomenon has been reported previously in vertebrates in part due to unmethylated CpG motifs within plasmid DNA molecules²⁷. Future studies with larger group sizes could be performed to investigate this possibility.”

Reviewer #3 (Remarks to the Author):

None of our concerns from the initial review were addressed in the revised manuscript.

1. The authors should acknowledge the previous published study that already demonstrated that in the absence of type I IFN signaling, Zika virus (ZIKV) can replicate in the testes and is found in the sperm. The distinctions between Govero et al. 2016 and the current study are minor differences that should be acknowledged in the introduction and not just in the response to the reviewers.

2. Because the data in Figures 1 and 2 are redundant of previously published work (see Govero et al. 2016), this paper should be focused on the value of the vaccine candidate. Showing that the vaccine reduces mortality and infection of reproductive tissues in males is

a start. But showing that the vaccine eliminates persistent infection (Reviewer 1) or transmission of virus to females (Reviewer 3) is necessary. Both requests were dismissed and without these additional, rigorous experiments, the data presented in the current manuscript are preliminary.

3. The relevance of extreme testicular pathology in the absence of type I IFN signaling in mice to the human condition should not just be addressed in the response to the reviewer, but should be detailed in the manuscript. What is provided as a response seems to be hearsay if nothing has been published documenting testicular pathology in humans, to date. There are no published data indicating that ZIKV infection makes males sterile, or there would be no sexual transmission. For this to be a relevant observation for vaccines or other therapeutics, there needs to be relevance to the human condition and this is not addressed, even as a limitation of the model, in the manuscript.

We agree with the reviewer that it would be beneficial to the readers if we were to include a detailed explanation of the differences between the experimental conditions used in the model of ZIKV-induced testicular atrophy described in this study in contrast to the work previously published by Govero et al. 2016 as well as other published studies. We have, therefore, included the following statements in the manuscript.

Lines 80-83:

“Further, exposure of male *Ifnar1*^{-/-} mice to ZIKV and exposure of type I IFN-depleted wildtype mice to mouse-adapted ZIKV results in severe damage to the testis, epididymis, and sperm with a measurable reduction in fertility^{17,18}.”

We agree that the ability to prevent persistent infection of the male reproductive tissues is an important feature of a protective vaccine against ZIKV. In our revised submission, we have included both testes weight and sperm parameter data at day 77 post-infection as Supplementary Data to show that the testes and sperm of vaccinated mice continue to be protected beyond 2 months after infection. We included the following statements in the manuscript.

Lines 193-195:

“At 77 d.p.i. the weight of the testes of unvaccinated and sham-vaccinated infected mice were significantly reduced compared to both mock-infected and prME-vaccinated infected mice (Supplementary Fig. 4).”

We also agree that further explanation of the relevance of these data to the human condition would be of interest to readers and would serve to clarify the limitations of using an immunocompromised mouse model for these studies. We have, therefore, included the following statements in the manuscript.

Lines 218-223:

“While the disease phenotype observed in ZIKV-infected *Ifnar1*^{-/-} mice, including weight loss and neurological signs, is more severe than the mild illness typically associated with human

infection; this model provides a stringent test to evaluate the ability of investigational therapeutics and vaccines to protect against ZIKV pathogenesis, including the mitigation and prevention of genitourinary signs and viral persistence within the male reproductive tract.”

Lines 234-245:

“ZIKV persistence in human semen^{7,8,9} and sperm¹⁰ has been documented for up to several months following the onset of symptoms, and pathological genitourinary symptoms have been described in infected men, including microhematospermia and dysuria^{12,13,28}. However, deleterious effects on sperm parameters and fertility, transient or otherwise, in ZIKV-infected males, including those exposed to the virus in utero, have not been reported to this day. Studies on this subject will be critical to better understand the physiopathology of ZIKV in humans as well as the full utility of the mouse models of ZIKV infection. There is, however, precedent for such an occurrence: while typically spontaneously self-resolving, bilateral mumps orchitis has been shown to result in suboptimal fertility in infected men²⁹. Further, sexual transmission of ZIKV between humans has been well documented^{7,11-14}, and additional studies are likely warranted in a mouse model of sexual ZIKV transmission³⁰ to also assess the ability of investigational therapeutics and vaccines to prevent this from occurring.”